# The Pleural Mesothelioma Cases and Mortality in Portugal in 2014–2020: A Descriptive Study

**DOI:** 10.3390/healthcare12111103

**Published:** 2024-05-28

**Authors:** Cátia Santos, Ema Sacadura-Leite, Joana Ferreira, Maria dos Anjos Dixe, Philippe Astoul, António Sousa-Uva

**Affiliations:** 1National School of Public Health, Public Health Research Centre, Comprehensive Health Research Center, NOVA University of Lisbon, 1600-560 Lisbon, Portugal; ema.leite@ulssm.min-saude.pt (E.S.-L.); asuva@ensp.unl.pt (A.S.-U.); 2Center for Innovative Care and Health Technology (ciTechCare), Polytechnic of Leiria, 2414-016 Leiria, Portugal; maria.dixe@ipleiria.pt; 3Occupational Health Department, Unidade Local de Saúde Santa Maria, 1649-028 Lisbon, Portugal; 4Union of Portuguese Misericordias, 1000-151 Lisbon, Portugal; joana_filipa_ferreira@hotmail.com; 5Department of Thoracic Oncology, Pleural Diseases, and Interventional Pulmonology, North Hospital, Aix-Marseille University, 13015 Marseille, France; philippejean.astoul@ap-hm.fr

**Keywords:** asbestos, exposure, malignant mesothelioma, pleural mesothelioma, incidence, mortality, years of life loss

## Abstract

Background: The incidence and mortality of pleural mesothelioma (PM) reflect the production and consumption of asbestos over time. However, despite the current global concern, these data remain to be known. Objective: Our aim was to carry out a descriptive analysis of PM cases and mortality from some Portuguese databases between 2014 and 2020. Methods: A retrospective observational study was carried out between 2014 and 2020. Data on the number of PM cases were provided by the Portuguese Cancer Registry, and data on mortality were from the Portuguese Death Certificate Information System. Results: Between 2014 and 2020, 315 cases of PM were reported, with 222 (70.5%) men. The average age of patients was 72.1, with the highest number of cases in patients aged >70 years (n = 198; 62.9%). The highest number of cases was reported in 2018 (n = 62; 19.7%). Regarding mortality, 169 deaths were reported, with 126 (74.6%) men and mostly in individuals aged >70 years (n = 109; 64.5%). It is estimated that around 520 years of potential life were lost. The highest number of deaths occurred in 2015 (n = 33; 19.5%). Conclusion: It is mandatory to reinforce the need for surveillance programs that allow us to gather real and reliable data and eliminate asbestos-related diseases.

## 1. Introduction

Pleural mesothelioma (PM) is a well-known rare cancer with a difficult diagnosis, poor prognosis, and a long latency period that mainly occurs as a result of exposure to asbestos [1]. The fact that PM mainly occurs due to exposure to asbestos fibers [1,2,3,4] and is the most common site of this disease [2], knowing its incidence and mortality data can lead to an accurate estimation of the consumption and production of asbestos in the past in industrialized countries [5,6] and a gross prediction of its impact in the future, taking into account the current consumption and production in developing countries.

Although some countries in recent years have begun to show real concern about the incidence and mortality of mesothelioma [7], at a global level, the data still remain scarce and unclear. This situation can be explained by the fact that PM is a rare disease [1] which can lead to significant variability in the diagnosis by health professionals [8]. Indeed, in only 69 countries [9], the use of asbestos has been banned, and this has happened at different times in each country [8]. Another explanation could be that categories C45 (malignant mesothelioma) and C45.0 (pleural mesothelioma) were only included in the International Classification of Diseases, 10th Revision (ICD-10) in 1993 which has not yet been implemented or fully implemented in many countries today [10].

Although much is still unknown, studies show that it is in northern and western Europe and Australia that the highest number of mesothelioma deaths occur [8]. Studies also show that the number of cases is beginning to fall in countries that have banned the use of asbestos before the 1990s, unlike those that banned its use after that date and experienced a peak in the 2000s [8].

Portugal, a country marked by a history of mining, is a country with a strong history of the consumption and production of asbestos, particularly in the construction industry [11]. Portugal joined the Asbestos Convention in 1999 [12], but it was only in 2005, in response to a European Union Directive, that it banned the use of chrysotile [9], a variety of asbestos believed to be less toxic [13,14]. It is estimated that around 90,605 tons of asbestos were used in the period 1920–1970 [15] and, between 1930 and 2003, around 113,131 tons [16]. In a study carried out between 2014 and 2016 in Portugal, it was found that 25% of 793 samples of buildings and equipment contained asbestos, namely of the chrysotile type [17]. In another study carried out on mesothelioma in the period of 2000 to 2011, 427 cases were identified. These data showed that in Portugal there is an estimated underreporting of 97 percent of cases as an occupational disease [18]. As in other countries, data in Portugal are unreliable.

Due to the importance of knowing the true impact of asbestos on public health in Portugal and trying to contribute to a global image actively, it is essential to develop surveillance programs to allow the establishment of an epidemiological evolution of PM in terms of incidence, survival, exposure, and medical–social recognition. Therefore, because only knowing the data makes it possible to act, this work aims to carry out a descriptive analysis of PM cases and mortality in some Portuguese databases between 2014 and 2020.

## 2. Materials and Methods

An observational and retrospective study was carried out between 2014 and 2020 to describe the number of cases of PM and its mortality in Portugal. The STROBE guidelines for observational studies [19] were followed.

Data were collected from 2 databases, namely data from the Portuguese Oncological Registry (RON) and from published annual death data on the website of the Death Certificate Information System (SICO). After analyzing the data, and due to some inconsistency in the mortality data between the different databases, the RON data were only considered for the number of cases and the SICO data for mortality.

As PM is a notifiable disease in Portugal that frequently requires hospitalization, data were also collected from the Social Security Institute (ISS) and the Portuguese National Database of Homogeneous Diagnostic Groups (GDH). However, due to the small number of reported cases of mesothelioma (23 cases) during the same period in the ISS database, and the impossibility of accessing data referring to patients (only for inpatient episodes) in the GDH database, ethical reasons and data reliability forced the exclusion of these data.

Only data from patients living in Portugal were included, and, knowing that only PM is monofactorial to asbestos exposure [2,4], only data referring to code C45.0 (pleural mesothelioma) of the ICD-10 were collected. All cases of PM were included, regardless of whether they were diagnosed in life or post mortem, provided they were morphologically proven. The study period, 2014–2020, was defined on the basis that only in this period was it possible to obtain data on PM (ICD-10 C45.0) in the two databases (RON and SICO).

From the RON, data were obtained for gender, age at diagnosis, district of residence, date of diagnosis, and morphology of the PM (ICD-10 morphology code M905, mesothelial neoplasms). From the SICO, data were obtained on the number of deaths (number of deaths in the resident population), the crude mortality rate (number of deaths observed during a given period per 100,000 inhabitants), and the potential years of life lost (number of years a given population theoretically loses if it dies prematurely—the sum of the products of deaths in each age group up to the age of 70 and the difference between the age of 70 and the average for each age group). For each of these, it was possible to extract already processed data relating to gender, age groups, region of residence, and place of death. To standardize the data, the RON data on age at diagnosis and district of residence were grouped into the age groups and regions defined by the SICO.

In addition, a literature search between 1965 and 2020 was carried out on PubMed, Web of Science, Google Scholar, and national and international registries to identify papers published in English, French, and Portuguese that could provide complementary information. However, due to the scarcity and heterogeneity of the information, only the data obtained from the RON and SICO were used.

Due to the nature of the data, only a descriptive analysis was carried out. Data compilation and descriptive statistical analyses were carried out using Microsoft Excel of Microsoft Office 365 Education^®^ (Microsoft Corporation, Redmond, WA, USA). This study was approved by the Ethics Committee of the National School of Public Health and by the Ethics Committee of the Porto Portuguese School of Oncology.

## 3. Results

Table 1 summarizes the PM cases and deaths reported to the RON and SICO, respectively, from 2014 to 2020.

### 3.1. Number of Cases of Pleural Mesothelioma

During this time, 315 (100%) patients were diagnosed with PM, where 222 (70.5%) were men and 93 (29.5%) were women. The average age at diagnosis was 72.1 years (standard deviation of 11.3 years), with a range of 33–96 years. Two hundred and fifty-two (80.0%) patients were diagnosed at the age of 65 or over, with the highest occurrence over an age of 70 (n = 198; 62.9%). Lisbon and Tagus Valley and the North were the regions with the most cases (n = 234; 74.3%), with the highest occurrence (n = 148; 47%) in Lisbon and Tagus Valley. Regarding morphology, of the 315 (100%) cases, 203 (64.4%) cases were coded as malignant mesothelioma, M 9050/3, and 99 (31.4%) cases as malignant epithelioid mesothelioma, M 9052/3. The years 2018 (n = 62; 19.7%) and 2014 (n = 54; 17.1%) had the highest number of cases (Figure 1).

### 3.2. Mortality of Pleural Mesothelioma

#### 3.2.1. Number of Deaths

During this period, there were 169 (100%) deaths, with 126 (74.6%) men and 43 (25.4%) women. One hundred and thirty-six (80.5%) deaths occurred at the age of 65 or over, with those over 70 having the highest number of deaths (n = 109; 64.5%). One death (0.6%) occurred for a woman in the 40–44 age group. Lisbon and Tagus Valley and the North were the regions with the highest number of deaths (n = 126; 74.6%), with Lisbon and Tagus Valley having the peak number of deaths (n = 85; 50.3%). One hundred and forty-three (84.6%) deaths occurred in a health unit. The highest number of deaths was recorded in 2015 (n = 33; 19.5%) and 2016 (n = 28; 16.6%) (Figure 2).

#### 3.2.2. Crude Mortality Rate

The crude mortality rate (×100.000 inhabitants) was 0.2 on average, with 0.4 for men and 0.1 for women. Regarding age groups, the rate varied on average between 0.1 (40–44 age group) and 1.0 (>70 age group). Regarding the region of residence, Alentejo, Algarve, Madeira Autonomous Region, and the Azores Autonomous Region were the regions with the highest crude death rates, with an average of 0.4, followed by Lisbon and Tagus Valley, with an average of 0.3.

#### 3.2.3. Potential Years of Life Lost

During this period, it is estimated that 520 potential years of life were lost, with 294 for men and 226 for women. It was in the 55–59 and 65–69 age groups that the highest number of years of potential life lost (242) was found; however, when analyzed by individual, it was in the 40–44 and 45–49 age groups that the highest numbers were seen, namely, 28 years in the 40–44 group and an average of 23 years per individual in the 45–49 group. Regarding the region of residence, Lisbon and Tagus Valley had the highest number of years of life potentially lost (234) in general. Still, considering the number of cases per region, the Azores Autonomous Region had the highest number of years of life potentially lost (23) per individual.

## 4. Discussion

### 4.1. Summary of Evidence

To carry out a descriptive analysis of the number of PM cases and mortality in Portugal, data were extracted from two different databases, namely, the RON and SICO. Mortality data could not be retrieved from the RON because of the 315 (100%) cases of PM, where only 201 (63.8%) patients had died (n = 125; 39.7% from PM and n = 76; 24.1% from other causes), and it was not possible to confirm whether the remaining 114 (36.2%) patients were still alive at the time of data collection. If true, these figures would imply that these patients had a survival rate longer than the average of 2 years estimated after a diagnosis of PM [20], which seems unbelievable.

Social–medical recognition is one of the aspects that is also becoming increasingly important when addressing this issue. Although these data were excluded for ethical and reliability reasons, it was found that only 11 mesotheliomas (pleural, pericardial, or peritoneal) were declared under the terms of the compulsory reporting of disease, and 23 patients were ISS beneficiaries for mesothelioma disease. Considering that in Portugal between 2000 and 2011, it was estimated that 97% of mesothelioma cases were underreported as an occupational disease [18], and if we consider the RON data for PM only (315 cases) and the ISS data, the underreporting figure may be even higher. No study was found in Portugal to support the causes of this underreporting.

In the period of 2014 to 2020, 315 (100%) cases of PM were reported to the RON. If we compare these data to those of the French National Mesothelioma Surveillance Program, which covers only 30% of its population [21], 1608 cases were reported in 2013–2017 [4], and the Italian National Registry (ReNaM), which reported 8868 cases in 1993–2004 [22]; taking into account the growing trend in the number of cases, the cases reported in Portugal seem to still be low compared to these industrialized countries.

The highest number of cases was observed in men and in patients over 65 years, which is in line with the literature [1,23]. The highest number of cases occurred in the Lisbon, Tagus Valley, and North regions, and while there are no data to support this information, it should only be noted that three of the large companies linked to asbestos, employing around 800 people, were in the Lisbon and Tagus Valley regions and another in the North [11]. On the other hand, the National Health Service (SNS) has the most differentiated diagnostic pulmonology and thoracic departments in those regions. Still, about diagnosis, more specifically morphology, of the PMs that could be pathology-classified (n = 112; 35.5%), 99 (31.4%) were epithelial, which is in line with the literature, which estimates that 70% of mesotheliomas are epithelial [24,25].

The years with the greatest number of cases were 2018 and 2014, and, from 2015 to 2018, there was an annual increase in cases. This increase can be explained by the fact that it is estimated that the peak of asbestos use in Portugal occurred in 1980 [26]. In this sense, and if we relate this to the latency period (20–50 years), the expected peak of PM will be between 2000 and 2030. If we take into consideration the fact that only in 2005 the use of all varieties of asbestos was banned in Portugal [9], we can say that, at least until 2025–2055, we will continue to have cases of PM. However, in 2019 and 2020, there was a decrease in the number of cases. Although the figures for 2019 contradict the upward trend seen for other cancers in 2020, it can be said that the reduction of around 20 percent in cases compared to 2018 may be related to the COVID-19 pandemic, which had a real impact on access to healthcare and therefore on diagnostics [27].

Regarding the number of deaths from PM, 169 (100%) were reported, with more in men over 65 years and for patients residing in the Lisbon and Tagus Valley regions. The data on age, gender, and residence are in line with those in the PM literature, while the number of deaths is far from the number of cases reported by the RON. In this sense, and if many of these people have already died, this situation could, eventually, be explained by the fact that these patients may have died from causes other than PM [24].

The crude mortality rate on average was higher for men, patients over 65 years, and those living in the Alentejo and Algarve regions, as well as the Madeira Autonomous Region and the Azores Autonomous Region. No study on PM was found to support these data, and, since no other tumor has the same characteristics as PM, particularly regarding the difficulty of diagnosis and low survival after diagnosis, the analysis of these data about other malignant tumors was also not considered.

Regarding potential years of life lost, no study directly related to PM was found. However, in a study carried out based on WHO data on mesothelioma in the period of 1994–2010, it was possible to find data supporting that although the number of potential years of life lost was higher in men, the average number of years lost was higher in women (related to average life expectancy) [28,29]. In fact, in our study, we found that 520 potential years of life were lost, with 294 in men and 226 in women, with an average of 2.3 years in men and 5.3 years in women.

The year with the most reported deaths was 2015. From 2016 to 2018, there was homogeneity in the number of deaths reported. In 2019 and 2020, there was a significant decrease in the number of cases. It is believed that, in 2019, this was due to delays in notification that were not corrected and that, in 2020, this was due to the COVID-19 pandemic, since COVID-19 represented the second leading cause of death, and there was a decrease in the number of deaths from malignant tumors [30].

The largest number of individuals died in a clinical care unit, a situation which, despite appearing to be changing with improved support, is still a reality that needs to change to respond to the needs and preferences of each individual [31].

The data presented allow us to state that it is mandatory to invest in surveillance programs that allow us to gather real and reliable data and consequently eliminate asbestos-related diseases in Portugal and around the world.

### 4.2. Implications for Future Research

Future research could examine the magnitude of the impact that asbestos exposure has on public health today and estimate the importance of banning it globally for the future, in addition to the importance of creating surveillance programs.

### 4.3. Limitations

The main limitations of this study are the scarcity of information on the occurrence and mortality of PM and the history of asbestos exposure, consumption, and production in both Portugal and worldwide. As well, another major limitation is the fact that there is no mesothelioma or asbestos exposure surveillance program in Portugal, which means that the data are scarce, decentralized in various databases, and not uniform.

## 5. Conclusions

It is recognized that the occurrence and mortality from PM reflect the massive use of asbestos in industrialized countries in the past and the current consumption and production in developing countries. In this sense, in the coming decades, this will continue, for some years, to be a serious global public health problem.

## Figures and Tables

**Figure 1 healthcare-12-01103-f001:**
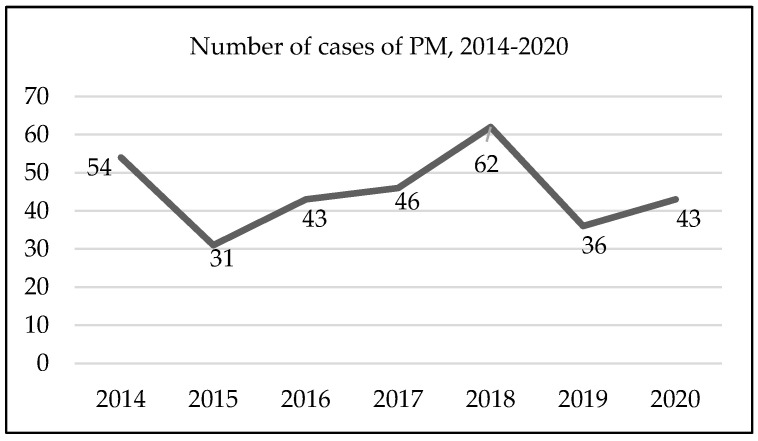
Number of cases of pleural mesothelioma reported to RON by year, 2014–2020.

**Figure 2 healthcare-12-01103-f002:**
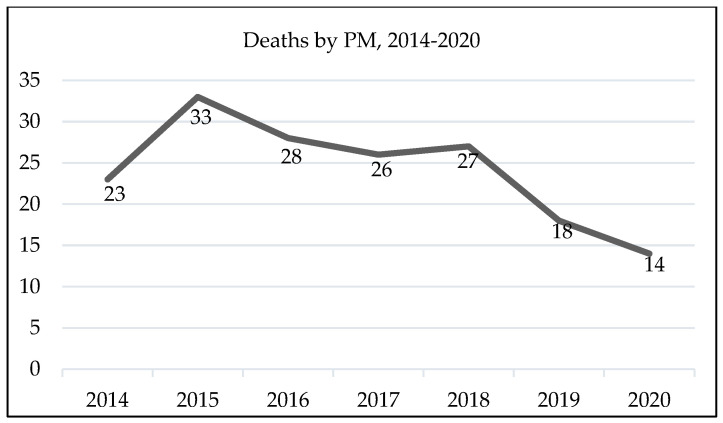
Number of deaths due to pleural mesothelioma reported to the SICO by year (2014–2020).

**Table 1 healthcare-12-01103-t001:** Pleural mesothelioma cases and deaths reported in the RON and SICO databases, 2014–2020.

Characteristics	Number of Cases *		Deaths **		Crude Mortality Rate **	Potential Years of Life Lost **
	No.	%	No.	%	Mean	No.
Total	315	100	169	100	0.2	520
Sex						
Male	222	70.5	126	74.6	0.4	294
Female	93	29.5	43	25.4	0.1	226
Age Groups						
30–34	1	0.3	NA ^a^	NA	NA ^a^	NA ^a^
35–39	0	0	NA ^a^	NA	NA ^a^	NA ^a^
40–44	2	0.6	1	0.6	0.1	28
45–49	1	0.3	5	3.0	0.2	115
50–54	6	2.0	3	1.8	0.2	54
55–59	22	7.0	10	5.9	0.2	130
60–64	31	9.8	14	8.3	0.3	112
65–69	54	17.1	27	15.9	0.6	81
>70	198	62.9	109	64.5	1.0	NA ^a^
Regions						
North	86	27.3	41	24.3	0.2	125
Central	42	13.3	19	11.2	0.2	32
Lisbon and Tagus Valley	148	47.0	85	50.3	0.3	234
Alentejo	21	6.7	9	5.3	0.4	37
Algarve	15	4.8	10	5.9	0.4	30
Madeira Autonomous Region	2	0.6	2	1.2	0.4	8
Azores Autonomous Region	0	0	1	0.6	0.4	23
Unknown	1	0.3	2	1.2	NA ^a^	NA ^a^
Place of death						
Health institution	NA ^a^	NA	143	84.6	NA ^a^	NA ^a^
Home	NA ^a^	NA	14	8.3	NA ^a^	NA ^a^
Other location	NA ^a^	NA	12	7.1	NA ^a^	NA ^a^
Morphology						
Malignant mesothelioma	203	64.4	NA ^a^	NA	NA ^a^	NA ^a^
Fibrous malignant mesothelioma	7	2.2	NA ^a^	NA	NA ^a^	NA ^a^
Malignant epithelioid mesothelioma	99	31.4	NA ^a^	NA	NA ^a^	NA ^a^
Biphasic malignant mesothelioma	6	2.0	NA ^a^	NA	NA ^a^	NA ^a^

*: RON Data; **: SICO data; ^a^: Data not available.

## Data Availability

Detailed data are available upon reasonable request to the corresponding author.

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
