# Peer review of "The Pleural Mesothelioma Cases and Mortality in Portugal in 2014–2020: A Descriptive Study"

_healthcare, 2024, doi:10.3390/healthcare12111103_

Round 1

Reviewer 1 Report

Comments and Suggestions for Authors

Dear Editor and Authors

 I read with interest the article by Catia Santos and colleagues entitled “Pleural Mesothelioma cases and mortality in a Western Euro- 2 pean country in 2014-2020: a descriptive study”.

Herein, authors conducted a retrospective observational study along 7 years by analyzing data of National cancer registry. The aim of the study was to take a picture of a terrible disease who typically affect patients with an exposure to asbestos.

The article is well-written with some minor issues and concerns.

Title should specify that this study is based on National (Portugal) data, and it represent a Portuguese picture more than a Western European analysis.  

Has the author included patients with a MPM diagnosed by autopsy?

In 19.5% MPM was diagnosed in patients younger than 65 years. How many of them were active workers? Have the authors evaluated also the economic impact of this aspect?

Moreover, is it expected in Portugal a kind a kind of insurance or financial compensation for these patients?

How the author explains the lack of data on the histology? there are data only on 99 epithelioid and 6 biphasic tumors.

Reviewer 2 Report

Comments and Suggestions for Authors

General comments:

A retrospective observational study of pleural mesoteliomas cases. The main objetive is a descriptive analysis of PM cases and mortality from some Portuguese databases between 2014 and 2020. 

This is a large series of cases of pleural mesothelioma. However, the descriptive analysis is quite simple. Perhaps this manuscript has more interest as a scientific letter than an original article.

The last sentence of the conclusions exceeds the objective of this manuscript. Reference could be made to this in the last part of the discussion.

Specifics comments:

.- Abstract. Delete the numbers “1,2…”

 .- Keywords. Maybe add the terms “malignant mesotelioma” and “years of life loss”

 .- References. 30 citations are presented, of which 8 (<25%) are recents, that is, five years or less from its publication. Consider including any recent additional references.

Comments on the Quality of English Language

It's ok

Reviewer 3 Report

Comments and Suggestions for Authors

Thank you for the opportunity to analyze this interesting analysis dealing with the mortality induced by a pleural mesothelioma in Portugal. 

Concerning the introduction:

            The introduction is very well written with and is also a good synthesis about major concerns about PM, moreover concerning the lack of data regarding the incidence and the mortality. No major concerns

Concerning the methodology:

            The two databases used for this study seems to be well selected regarding the incidence, analyzed with the RON data and the mortality with the SICO data, taking account “the real-life utilization of these databases” 

            That’s quite a surprise to notice that only 23 cases of PM were registered in the ISS, do you have an explanation? 

            Regarding the chosen-period, why did you start this analysis in 2014? And not earlier? 2005 for example? 

Concerning the results

            Results are well reported and clearly presented in the tables, no concerns. 

Concerning the discussion:

It’s a well written discussion very well documented with good references. 

            “the real-life utilization of databases”: A true challenge. The underreporting of PM is still quite a surprise, because, economic considerations are engaged with PM. Do you have an explanation? Lack of information? Lack of interest, fatality? Or something else? 

            As it’s mentioned, there may be a link between the higher incidence of PM and the area of PM declaration due to the presence of the main asbestos factories. But hard to prove it.

            Concerning the incidence, how many patients were in a “kind of screening program of PM post asbestos exposure” , because it doesn’t exist in Portugal as you have mentioned it. 

Limitations are well described. 

Concerning the conclusion:

            Clear.

            It’s a well written, easy reading and interesting article, about still a burden subject that is accepted with a supplemental information’s about my comments. 

            Congratulations to authors for this work. 

Round 2

Reviewer 2 Report

Comments and Suggestions for Authors

The changes suggested by the Reviewers Team are positively appreciated. Congratulations!